# INFORMATION REGULARIZED NEURAL NETWORKS

## ABSTRACT

We formulate an information-based optimization problem for supervised classification. For invertible neural networks, the control of these information terms is passed down to the latent features and parameter matrix in the last fully connected layer, given that mutual information is invariant under invertible map. We propose an objective function and prove that it solves the optimization problem. Our framework allows us to learn latent features in a more interpretable form while improving the classification performance. We perform extensive quantitative and qualitative experiments in comparison with the existing state-of-the-art classification models.

## 1 INTRODUCTION

Quantities of information are nonlinear measures capable of describing complex relationship between unstructured data and they form the basis of the probabilistic algorithms in the literature of machine learning. Information theoretic methods are also reported to be effective on improving deep generative models (Chen et al. (2016); Kim & Mnih (2018)) and deep learning models for classification (Grandvalet & Bengio (2004); Pereyra et al. (2017)). Information Bottleneck (IB) problem (Tishby et al., 1999) is formulated as:

$$minimize \ \mathrm{I}(X;T) - \lambda \mathrm{I}(Y;T) \ , \tag{1}$$

where the solution random variable $T$ is interpreted as a minimal sufficient representation of signal $X$ for label $Y$ and the mutual information is defined as

$$\mathrm{I}(X;Y) = \int_{\mathcal{X}} \int_{\mathcal{Y}} p(x,y) \log \left( \frac{p(x,y)}{p(x)p(y)} \right) dy dx \ . \tag{2}$$

The term $\mathrm{I}(X;T)$ has its origins in Lossy Compression and Rate-Distortion Theory (Cover & Thomas, 2006), conveying an simple idea of "keep only what is relevant".

However, Saxe et al. (2018) argued that the mutual information $\mathrm{I}(X;T)$ between signal $X$ and feature $T$ in intermediate layer is infinite, as the transformation from $X$ to continuous random variable $T$ is deterministic. In addition they showed experimentally that layers equipped with ReLU actually do not compress too much information, which is supported by many recent work on the invertibility of the neural network (Dosovitskiy & Brox (2015); Jacobsen et al. (2018)). This motivates us to consider a different problem with similar principle idea: we would like to establish a theoretically valid objective that allows the neural network to extract only the relevant information for classification from the data.

We focus on the discrete prediction random variable $\widehat{Y}$ inferred by the probabilistic model $\mathbb{P}(\widehat{Y}|X)$ and introduce the following information optimization problem for supervised classification:

$$\begin{aligned} &maximize \ \mathrm{I}(Y;\widehat{Y}) \\ &subject \ to \ \mathrm{I}(X;\widehat{Y}) - \mathrm{I}(Y;\widehat{Y}) < \tau \ , \text{ for some } \tau > 0 \ . \end{aligned} \tag{3}$$

The intuition behind this objective lies in twofold:

**Information perspective**: A good classification model should be robust against irrelevant features of $X$, and prevent over-fitting in the learning process. In optimization problem (3) we maximize the relevant information $\mathrm{I}(Y;\widehat{Y})$, while constraining the irrelevant information $\mathrm{I}(X;\widehat{Y}) - \mathrm{I}(Y;\widehat{Y})$

that $X$ has about $\widehat{Y}$. Although $\mathrm{I}(X;\widehat{Y}) - \mathrm{I}(Y;\widehat{Y})$ converges to zero as $\mathrm{I}(Y;\widehat{Y})$ approaching its maximum (see Figure 1L), in practice it's never attained due to the limited capacity of the models or over-fitting. Our proposed constrain addresses the problem of over-fitting: if two models achieve the similar classification accuracy, this constraint prefers the one that does not overfit to spurious factors of variation in $X$ (e.g., pixel-level artifact/noise in the image that accidentally correlates to the labels in the training data).

**Prediction confidence perspective**: A good classification model should not be certain about its decision which is in fact wrong. However, modern neural networks are too confident in their predictions (Guo et al., 2017; Szegedy et al., 2015; Pereyra et al., 2017). To be more precise, high capacity neural networks mostly assign labels of data with prediction confidence near 0 or 1. In particular, they assign 0 probability to some correct labels and therefore do not have enough flexibility to correct themselves from making the wrong prediction. We propose to compress the irrelevant information $\mathrm{I}(X;\widehat{Y}) - \mathrm{I}(Y;\widehat{Y})$, where minimizing $\mathrm{I}(X;\widehat{Y})$ decreases the confidence on all predictions but maximizing $\mathrm{I}(Y;\widehat{Y})$ increases the confidence on the correct predictions. Therefore the overall effect reduces the certainty on the false prediction of $\widehat{Y}$ (see Figure 1L).

To solve this optimization problem, we first present some insights on the dynamics of deep neural network, which can be decomposed into two stages: (i) Transformation stage: samples $\{X_k\}_{k=1:n}$ of the high dimensional unstructured signal $X$ are transformed under the deep invertible (information preserving) feature map $F$ to become (almost) linearly separable; (ii) Classification stage: the weight matrix $w$ in the last fully connected layer together with the Softmax function, takes structured features $\{F(X_k)\}_{k=1:n}$ as input and gives predictions.

Invertibility of $F$ allows neural networks to pass the control of $\mathrm{I}(X;\widehat{Y}) = \mathrm{I}(F(X);\widehat{Y})$ towards $F(X)$ and $w$ in the last layer, where $F(X)$ can be interpreted as transformed signal that preserves information about the original signal $X$ and the inference model becomes conceptually linear with classifier $w$ (see Figure 1R). In Section 2 we derive objective function (7) and prove that it solves the optimization problem (3). We show the classification performance is improved in Section 4.1 and the features $F(X)$ are sculpted into a form with more interpretability entry-wise in Section 4.2.

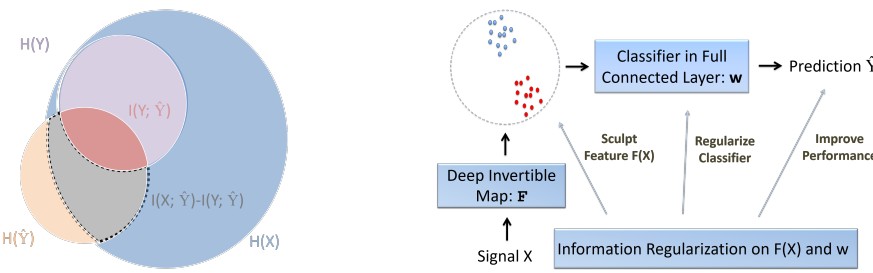

Figure 1: (L) An information Venn diagram: three disks represent the entropy of $X, Y, \widehat{Y}$ respectively, the area in red is the relevant information $\mathrm{I}(Y;\widehat{Y})$, the area in grey is the irrelevant information $\mathrm{I}(X;\widehat{Y}) - \mathrm{I}(Y;\widehat{Y})$. The optimal solution is obtained when the smaller disks coincide, which is typically not achieved in practice. In particular, the trained model may be extremely confident in its prediction (when $H(\widehat{Y})$ lies inside of $H(X)$), but predicts the wrong label (having large grey area). Our optimization problem explicitly prohibits the growth of grey area throughout the training. (R) Logic chart of our formulation: our proposed optimization problem only involves $F(X)$ and $w$, allowing us to have control over the latent feature $F(X)$ directly.

The invertibility property has been empirically demonstrated for complex non-linear deep neural networks that are widely used in practice. We will discuss the literature in Section 3. In addition, we prove in Proposition C.1 that a lower bound of classification error is minimized if neural network is invertible.

**Our contribution:** Our contribution lies in the following: (i) we formulated a novel information optimization problem for supervised classification; (ii) we propose a simple objective function that improves supervised deep learning with better performance and interpretability; (iii) we formally

justify the use of $\ell_1, \ell_2$ regularization from an information perspective. Different from the naive regularization on $w$, our regularization on $w^T F(X)$ is novel and effective.

## 2  MAIN RESULTS

Consider a classification problem where the training data $\mathcal{D} = \{(x_k, y_k)\}_{k=1:n}$ are sampled from random variable pair $(X, Y)$ with unknown joint distribution. Each $x_k$ is fed into a deep probabilistic model, which outputs probability densities and predicts $\widehat{y}_k$, a realization of the prediction random variable $\widehat{Y}$. Let $\mathcal{C}$ denote the label class and $\mathcal{X}$ denote the signal space, then the mutual information between random variables, e.g., continuous $X$ and discrete $\widehat{Y}$, is defined as $\mathrm{I}(X; \widehat{Y}) = \sum_{\widehat{y} \in \mathcal{C}} \int_{\mathcal{X}} p(x, \widehat{y}) \log\left(\frac{p(x, \widehat{y})}{p(x)p(\widehat{y})}\right) dx$, and the entropy of $Y$ is defined as $H(Y) = -\sum_{y \in \mathcal{C}} p(y) \log p(y)$.

We first call out our assumptions used throughout our analysis. **(I)**: we assume the marginal densities of $Y, \widehat{Y}$ are uniform over $\mathcal{C}$; **(II)**: there exists a unique true label for every sample of $X$.

Mutual information is bounded and its gradient with respect to logits is approximately zero over a large domain. In particular if the logits are initially small for true labels, gradient updates cannot effectively correct them. Therefore it's not practical to train mutual information terms directly. In this section we introduce alternative terms and prove that they are feasible for our purpose. We show in Proposition 2.2 that $\mathrm{I}(Y; \widehat{Y})$ is maximized if the classical cross entropy objective is minimized. On the other hand, we show in Proposition 2.1 that for invertible $F$, $\mathrm{I}(X; \widehat{Y}) = \mathrm{I}(F(X); \widehat{Y})$ is minimized if $\|w^T F(X)\|$ is minimized. We derive our objective function (7) in Section 2.2. Our experimental result in Section 4.1 verifies that the proposed objective function does compress the irrelevant information $\mathrm{I}(X; \widehat{Y}) - \mathrm{I}(Y; \widehat{Y})$.

### 2.1  SOLVING OPTIMIZATION PROBLEM WITH FEASIBLE TERMS

Without loss of generality, we consider the binary classification problem, i.e. the label class $\mathcal{C} = \{\pm 1\}$. To tract the population quantities $\mathrm{I}(F(X); Y)$ and $\mathrm{I}(Y; \widehat{Y})$, we decompose each of them into an empirical part and a probabilistic bound, which is negligible if sample size $n$ is large. In Proposition 2.1, we show that in order to compress $\mathrm{I}(F(X); Y)$, we need to compress the norm of classifier $w$ and feature $F(X)$. In particular, smaller $|w^T F(X)|$ represents lower confidence of the model on its predictions $\widehat{Y}$, indicating a smaller amount of mutual information $\mathrm{I}(F(X); \widehat{Y})$. The proof is provided in Appendix A.

**Proposition 2.1.** $\mathrm{I}(X; \widehat{Y}) = \mathrm{I}(F(X); Y)$ *is well estimated by its empirical version* $\sum_{k=1}^{n} \sum_{\widehat{y} \in \mathcal{C}} p(\widehat{y}|x_k) \log(2p(\widehat{y}|x_k))/n$ *with high probability, which shares the same unique (global) minimum with* $\sum_{k=1}^{n} |w^T F(x_k)|$ *at* $w^T F(x_k) = 0$, *for all* $k \in \{1, ..., n\}$.

Denote the sigmoid function with $\sigma(a) = 1/(1 + e^{-a})$. Proposition 2.2 establishes the relationship between maximization over mutual information $\mathrm{I}(Y; \widehat{Y})$ and minimization over cross entropy $-\sum_{k=1}^{n} \log \sigma(y_k w^T F(x_k))$; higher confidence of the model on its correct predictions over the samples indicates a larger value of $\mathrm{I}(Y; \widehat{Y})$. The proof is provided in Appendix B.

**Proposition 2.2.** $\mathrm{I}(Y; \widehat{Y})$ *is well estimated by its empirical version* $\sum_{y\widehat{y}} \widehat{\pi}_{y\widehat{y}} \log\left(\frac{\widehat{\pi}_{y\widehat{y}}}{\widehat{\pi}_{y+} \widehat{\pi}_{+\widehat{y}}}\right)$ *with high probability, which shares the same unique (global) maximum with* $\sum_{k=1}^{n} \log \sigma(y_k w^T F(x_k))$ *given that* $y_k w^T F(x_k) > \frac{1}{2}$, *for all* $k \in \{1, ..., n\}$. *Here* $\widehat{\pi}_{y\widehat{y}} = \frac{1}{2n_y} \sum_{i=1}^{n_y} \sigma(\widehat{y} w^T F(x_i))$ *is an unbiased estimate of* $p(y, \widehat{y})$ *and* $\widehat{\pi}_{y+} = \sum_{\widehat{y} \in \mathcal{C}} \widehat{\pi}_{y\widehat{y}}, \widehat{\pi}_{+\widehat{y}} = \sum_{y \in \mathcal{C}} \widehat{\pi}_{y\widehat{y}}$.

### 2.2  DERIVATION OF OBJECTIVE FUNCTION

In Lagrangian form of optimization problem (3), the constant $\tau$ can be dropped and the objective becomes

$$\textit{maximize } (1 + \lambda)\mathrm{I}(Y; \widehat{Y}) - \lambda \mathrm{I}(F(X); \widehat{Y}) \iff \textit{maximize } \mathrm{I}(Y; \widehat{Y}) - \frac{\lambda}{1 + \lambda}\mathrm{I}(F(X); \widehat{Y}) . \quad (4)$$

Consider a single signal $x_k$ and its true label $y_k$, we propose the following objective function for binary supervised classification problem:

$$\mathcal{L}_k = \alpha \mathcal{R}\left(|w^T F(x_k)|\right) - \log \sigma(y_k w^T F(x_k)) \, , \tag{5}$$

where $\mathcal{R}$ is some regularizer. According to results in Section 2.1, minimizing (5) allows us to maximize $\mathrm{I}(Y; \widehat{Y})$ while constraining $\mathrm{I}(X; \widehat{Y})$. We typically choose the hyper-parameter $\alpha > 0$ to be a reasonably small number. The intuition comes from the observation that $\lambda/(1 + \lambda)$ is upper bounded by one. If we compress $\mathrm{I}(X; \widehat{Y})$ harshly, then neural networks may choose to minimize $\mathrm{I}(F(X); \widehat{Y})$ at a cost of minimizing $\mathrm{I}(Y; \widehat{Y})$.

Recall from the information theoretic perspective of our proposed optimization problem, our regularizer should prefer a model that does not overfit among all the ones with high training accuracy. In this case neural networks assign only large logits $w_{y_k}^T F(x_k)$ to true label $y_k$ for each signal $x_k$, and generalization of (5) to multi-class case for $\mathrm{I}(F(X); \widehat{Y})$ can be simplified to constraining $w_{y_k}^T F(x_k) - w_j^T F(x_k)$, where $w_j$ is the $j$th column of $w$, assigning feature $F(x_k)$ a probability to label $j$. We propose to simply constrain $w_{y_k}^T F(x_k)$ and does not encourage increasing $w_j^T F(x_k)$:

$$\mathcal{L}_k = \alpha \mathcal{R}\left(|w_{y_k}^T F(x_k)|\right) - \log\left(\frac{e^{w_{y_k}^T F(x_k)}}{e^{w_{y_k}^T F(x_k)} + \sum_{j \neq y_k} e^{w_j^T F(x_k)}}\right) \, . \tag{6}$$

In our experiment, we take the Elastic Net approach by Zou & Hastie (2005) using a combination of $\ell_2$ and $\ell_1$ regularizers: we use Holder's inequality to bound $|w_{y_k}^T F(x_k)|$ with both $\sup F(X) \cdot \|w_{y_k}\|_1$ and $\|w_{y_k}\|_2 \cdot \|F(x_k)\|_2$. In practice we assume $\sup F(X)$ to be a constant and is absorbed into the hyper-parameter. Our proposed objective function is of the form:

$$\mathcal{L} = \alpha_1 \|w\|_1 + \frac{1}{n} \sum_{k=1}^{n} \left\{ \alpha_2 \|w_{y_k}\|_2 \cdot \|F(x_k)\|_2 - \log\left(\frac{e^{w_{y_k}^T F(x_k)}}{e^{w_{y_k}^T F(x_k)} + \sum_{j \neq y_k} e^{w_j^T F(x_k)}}\right) \right\} \, . \tag{7}$$

Notice that classical training methods maximize cross entropy and therefore $\mathrm{I}(Y; \widehat{Y})$, but do not compress $\mathrm{I}(X; \widehat{Y})$ explicitly. In Equation (7), we explicitly encourage the compression of irrelevant information $\mathrm{I}(X; \widehat{Y}) - \mathrm{I}(Y; \widehat{Y})$. As we demonstrate in Section 4.3, the proposed objective function does encourage a smaller amount of $\mathrm{I}(X; \widehat{Y}) - \mathrm{I}(Y; \widehat{Y})$ throughout the training process.

## 3 RELATED WORK

Recent experimental work reported that neural networks with invertible structure have better performance. Dosovitskiy & Brox (2015) showed that images can be resconstructed from the latent features in AlexNet through an inverting process; this reconstruction is further improved by Zhang et al. (2016), where they built an encoder-decoder structure to encourage invertibility and showed reconstructive objective is beneficial to the performance of the neural network (e.g., VGGNet). Shang et al. (2016) proposed an invertible activation scheme named CReLU to preserve information; Gilbert et al. (2017) analyzed theoretically the invertibility of CNN; Jacobsen et al. (2018) built a theoretic invertible structure whose performance is comparable to ResNet He et al. (2015). Invertibility seems to be an intriguing property or design principle that often emerges in the recent state-of-the-art deep architectures.

Information theoretic methods are reported to be effective attacking machine learning problems. In deep learning, IB was first introduced in Tishby & Zaslavsky (2015) and the follow-up experimental work Shwartz-Ziv & Tishby (2017). They argued that DNN structure forms a markov chain and information is compressed layer by layer. The theoretical breakthrough by Achille & Soatto (2017) established a connection between the IB objective and the generalization in deep learning, carrying along with the notion of sufficiency and invariance of representations. Strouse & Schwab (2016),Slonim & Tishby (1999) used IB objectives for clustering problems. Alemi et al. (2017), Gao et al. (2018),Kim & Mnih (2018) established information theoretic approaches based on VAE to encourage disentangled and informative latent representations; Grandvalet & Bengio (2004) introduced entropy regularizer in semi-supervised learning; Krause et al. (2010) took Regularized

| NaiveReg ResNet-32 | $\alpha_2 = 0.0005$ | $\alpha_2 = 0.001$ | $\alpha_2 = 0.002$ | $\alpha_2 = 0.004$ |
|---|---|---|---|---|
| Best Accuracy % | $70.06 \pm 0.38$ | $69.94 \pm 0.33$ | $69.86 \pm 0.32$ | $68.99 \pm 0.52$ |

| ResNet-32 | Original | $\alpha_2 = 0.01$ | $\alpha_2 = 0.03$ | $\alpha_2 = 0.05$ |
|---|---|---|---|---|
| Best Accuracy % | $70.15 \pm 0.33$ | $70.34 \pm 0.27$ | $\mathbf{70.57 \pm 0.20}$ | $70.25 \pm 0.23$ |
| Constrain | $0.296 \pm 0.044$ | $0.293 \pm 0.043$ | $0.280 \pm 0.040$ | $\mathbf{0.266 \pm 0.038}$ |

| ResNet-Wide | Original | $\alpha_2 = 0.01$ | $\alpha_2 = 0.05$ | $\alpha_2 = 0.09$ | $\alpha_2 = 0.15$ |
|---|---|---|---|---|---|
| Best Accuracy % | $78.51 \pm 0.27$ | $79.37 \pm 0.18$ | $79.62 \pm 0.13$ | $\mathbf{79.64 \pm 0.12}$ | $79.45 \pm 0.14$ |
| Constrain | $0.254 \pm 0.056$ | $0.240 \pm 0.051$ | $0.213 \pm 0.049$ | $0.198 \pm 0.044$ | $\mathbf{0.174 \pm 0.038}$ |

Table 1: Performance comparison on CIFAR100, Best Accuracy (%, test set) and the average values of the constrain $I(X; Y) - I(Y; \widehat{Y})$ throughout the training process are provided in the tables. The first table gives the results of naive regularization on $w$ for ResNet-32, with $\alpha_1 = 0, \alpha_2 = [0.0005, 0.001, 0.002, 0.003, 0.004]$, note that the choices of hyper-parameters are small because it regularize the whole matrix $w$ but not to specific columns as proposed; the second table is for ResNet-32, comparing original ResNet and RegResNets with $\alpha_1 = 0, \alpha_2 = [0.01, 0.03, 0.05]$; the third table is for ResNet-28-10-Wide, comparing original ResNet and RegResNets with $\alpha_1 = 5e^{-6}, \alpha_2 = [0.01, 0.05, 0.09, 0.15]$. All results are calculated from 10 samples. The constrain decreases as $\alpha_2$ increases; the performance can be effectively improved under proper choice of $\alpha_2$.

Information Maximization(RIM) approach for classification problems; Chen et al. (2016) proposed to add information ingredients to the objective of GANs, encourage to learn disentangled representations.

Our framework decomposes deep neural network into a composition of nonlinear transformation map and a linear probablistic model; this idea was originated in Bell & Sejnowski (1995) where they considered the blind separation problem and decompose the prediction $\widehat{Y}$ to be the sum of an invertible deterministic part and a stochastic part. Amjad & Geiger (2018) and Kolchinsky et al. (2017) also studied the IB problem in a stochastic setting. Our idea of explicit regularization on $w$ and $F(X)$ is related to the margin based and stability based interpretations of generalization in deep learning respectively, studied by Arora et al. (2018), Bartlett & Mendelson (2003), Neyshabur et al. (2017), Sun et al. (2015).

## 4 EXPERIMENTS

In our experiments we build the feature map $F$ with ResNet or InvNet (introduced in Section 4.2). In Appendix D we prove that ResNet by He et al. (2015) is invertible under mild assumptions. We prefix the name of the model trained under our objective with "Reg", i.e. RegResNet/RegInvNet.

### 4.1 PERFORMANCE

We report the accuracy of ResNet on test data of CIFAR100 in Table 1.[1]

We compare the performance between our proposed regularization on $w_{y_k}^T F(x_k)$ and the naive regularization on $w$. In both form of regularization we take $\alpha_1 = 0$ and test over different choices of $\alpha_2$. We pick smaller $\alpha_2$ for naive regularization because it's applied to the full matrix. We observe the performance of ResNet-32 under naive regularization drops monotonically as $\alpha_2$ increases.

Under a suitable choice of hyperparameters, RegResNet outperforms ResNet by a noticeable margin. It implies that our proposed constrain on the irrelevant information $I(X; \widehat{Y}) - I(Y; \widehat{Y})$ is beneficial to the classification performance. However, if the hyperparameters are too large, the performance drops, i.e. $\alpha_2 = 0.05$ for ResNet-32 and $\alpha_2 = 0.15$ for ResNet-Wide; this matches the discussion in Section 2.2 that the model may try to reduce the relevant information $I(Y; \widehat{Y})$. Our approach addresses the problem of over-fitting; ResNet-Wide is improved by a larger margin compared to

---

[1] We use the open source code implemented by Xin Pan at `https://github.com/tensorflow/models/tree/master/research/resnet` and build our objective function based upon it. Our results for the baseline models match the reported ones on the website.

ResNet-32 because ResNet-Wide has higher capacity and is therefore over-fits more to the training data.

## 4.2 ANALYSIS ON FEATURES

We introduce another invertible structured neural network on MNIST dataset and analyze the feature $F(X)$ learned in the last layer qualitatively. The feature map $F$ is built to be LeNet-300-100 and the decoder $D$ has the opposite structure. At each step during the training process, we update the autoencoder $F + D$ and the InvNet $F + w$ alternatively; our regularization is applied to $w$ as usual. In this section we report our result with $\alpha_1 = \alpha_2 = 0.002$.

### 4.2.1 LEARNING TO OVERLOOK IRRELEVANT INFORMATION

We feed 1k testing samples of digit 9 into the neural network to get 1k samples of features $F(X)$. Recall that the features $F(X)$ are vectors of dimension 100 and $w$ is a matrix of size $100 \times 10$. We calculate the mean and standard deviation of each entry of the features from these 1k samples. From Figure 2 we see that under our regularization, the entry-wise products $w_{10i}^T F(X)_i$ becomes sparse as only a few entries have large values for both feature and weight. This implies that the information needed to compute the logits for classification is encoded into only a few entries of the feature, which we regard as relevant entries. Our regularization forces small $w_{10}$ on irrelevant entries, so the logits it outputs are not sensitive to variations in these entries; on the other hand, we do encourage large $w_{10}$ on the relevant entries. This matches our motivation that a neural network should be robust against irrelevant information $I(F(X); \widehat{Y}) - I(Y; \widehat{Y})$ and focus on the relevant information $I(Y; \widehat{Y})$.

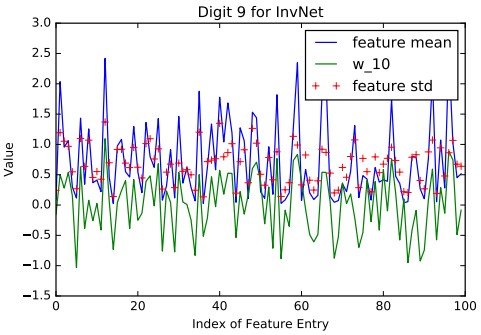 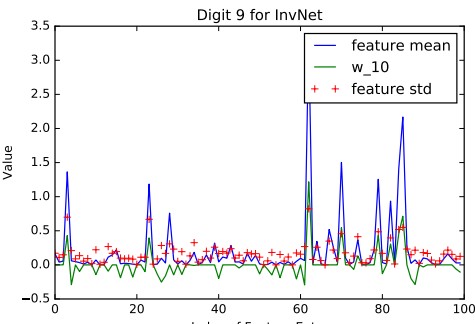

Figure 2: Statistics plots for feature entries of digit 9. Recall that the features $F(X)$ are vectors of dimension 100 and $w$ is a matrix of size $100 \times 10$. The horizontal coordinate represents the index of entries of $F(X)$ ranging from 0 to 99, the vertical coordinate represents entry values of the 10th column of $w$, the values of mean and standard deviation for each entry of features from samples of digit 9. For RegInvNet, $w_{10}$ only assigns large value to representative entries and is more robust against perturbations in other irrelevant entries.

### 4.2.2 LEARNING TO EXTRACT RELEVANT INFORMATION

It had been argued by Szegedy et al. (2013) that it is the space but not individual units in high level features that encodes interpretable information. Under our regularization, a meaningful basis of the informative space is found; in particular, the features of 9 are encoded into 10 entries (see Figure 2). On the other hand, features that have high values in these entries are expected to be the features of digit 9.

To validate this conjecture, for each digit, we find the entry of its feature mean with the highest value. We use the micro and macro average ROC metric on these feature entries and compare the results from InvNet and RegInvNet in Figure 3. The curve with larger area underneath indicates higher representative power of individual entries learned in the features. We conclude that under our regularization, relevant information for classification is encoded into only a few key entries of the features, and these entries are highly indicative and interpretable.

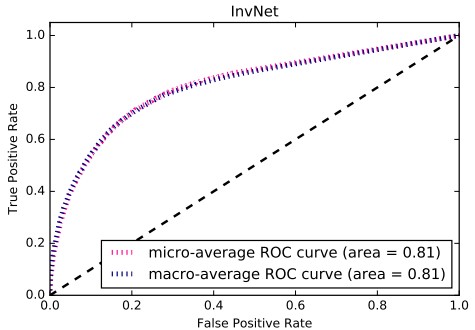
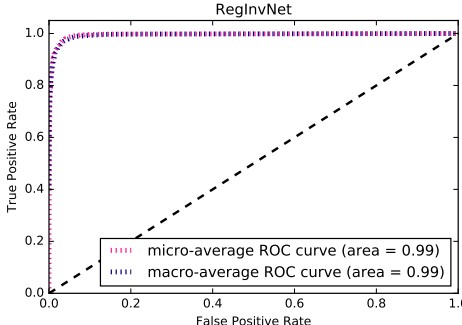

Figure 3: The macro/micro average ROC curves for representative entries of features genearated by original and regularized model. The entries of features learned under our regularization strongly indicate the categories of the digits.

### 4.3 RELATIONSHIP TO LINEAR REGRESSION

We trained ResNet-32 on CIFAR10, the feature $F(X)$ is a vector of size 64, the classifier $w$ is a matrix of size $64 \times 10$. Note that the product $w^T F(X)$ is a vector of size 10 representing the probability assigned by the model for each class.

Under our framework, deep learning can be conceptually simplified to regularized linear regression if we regard $F(X)$ as input data. However, $F(X)$ depends on the model parameters in the previous layers so it's not fixed like real data. Moreover we observe in our experiments that naive regularization on $w$ alone will upscale the norm of $F(X)$, which neutralizes the effect of regularization. In Figure 4 we show that as under our regularization, the $\ell_2$-norm of $w$ is suppressed while the $\ell_2$-norms of feature $F(X)$ remain similar. In addition, several rows of $w$ are trained to be zero, which implies that many entries of the feature $F(X)$ are regarded by the network as irrelevant information for classification, since any variance in the entries of $F(X)$ where the corresponding rows of $w$ are zero has no influence on the probability assigned to each label class by the model.

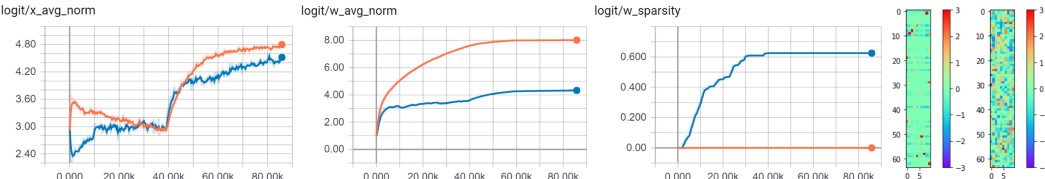

Figure 4: Compression of the RegResNet-32 (Blue) and the original ResNet-32 (Orange) on CIFAR-10 over the training process: the first plot records the average $\ell_2$ norm of the last layer features $F(X)$ in a batch; the second plot records the average $\ell_2$ norm of the columns of $w$; the third plot records the ratio of zero entries among all entries of $w$; the plots for trained $w$ with/without regularization after 84000 steps are listed on the right. Best test accuracy are 92.86% and 92.64% for regularized and original ResNet-32 respectively. Under our regularization, the norm of the feature learned remains similar and the norm of classifier $w$ is smaller. Therefore $w$ is less sensitive to "support" and "outlier" features.

## 5 THE ROLE OF INVERTIBILITY

Invertibility allows us to treat $F(X)$ as transformed data that preserves all the information from $X$, and therefore work on the information regularization problem under a linear scheme.

In Appendix D we prove that ResNet is fairly invertible due to the intrinsic invertibility of the operator $I + \mathcal{L}$ given $\|\mathcal{L}\| < 1$. In this section we build a PlainNet by using only $\mathcal{L}$ as the operator for each building block, so the theoretical guarantee for invertibility is not present for PlainNet. In Table 2,

we see that PlainNet-32 can still benefit from our regularization, however, it's performance is less stable compared to ResNet-32 if the hyper-parameters are too large. The reason is for PlainNet, the feature in the last layer $F(X)$ does not preserve information about $X$ very well, so it has a higher demand on the capacity of the classifer $w$ and is therefore more sensitive to our regularization.

| Performance % | Original | $\alpha_2 = 0.01$ | $\alpha_2 = 0.05$ | $\alpha_2 = 0.09$ | $\alpha_2 = 0.15$ | $\alpha_2 = 0.3$ |
|---|---|---|---|---|---|---|
| ResNet-32 | $92.49 \pm 0.14$ | $92.52 \pm 0.30$ | $92.76 \pm 0.33$ | $92.45 \pm 0.37$ | $88.36 \pm 3.12$ | $78.95 \pm 4.33$ |
| PlainNet-32 | $90.06 \pm 0.21$ | $90.33 \pm 0.24$ | $90.25 \pm 0.24$ | $90.06 \pm 0.31$ | $85.97 \pm 3.40$ | $62.76 \pm 16.22$ |

Table 2: The performance statistics for ResNet-32 and PlainNet-32 without or under various regularizations. For regularized models we hold $\alpha_1 = 1e^{-5}$ and take $\alpha_2 = [0.01, 0.05, 0.09, 0.15, 0.3]$. The means and standard deviations reported are based on 5 samples. It can be observed that although PlainNet-32 obtains marginal improvement for small $\alpha_2$, the performance drops dramatically and becomes unstable as $\alpha_2$ increases.

## 6 CONCLUSION

We give an interpretation of the deep learning dynamics by decomposing it into an signal transformation stage and feature classification stage, where we emphasis importance of the classifier $w$ in the last fully connected layer given that the feature map $F$ is invertible. Then we take the advantage of the fact that mutual information quantities are invariance under invertible mapping to attack our proposed information optimization problem for supervised classification in deep learning. Our theory justifies the use of direct regularization terms on $w, F(X)$ for neural networks with invertibility property. Our regularization improves the performance of neural networks by a noticeable margin and is capable of encouraging the interpretability of the entries of features learned in the last layer.

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

## A  PROOF OF PROPOSITION 2.1

Proposition 2.1 establishes a connection between $\mathrm{I}(X, \widehat{Y})$ and the absolute value of the logits $|w^T F(X)|$ for the binary case. Intuitively, decreasing the confidence of the model on its predictions will decrease the mutual information $\mathrm{I}(X; \widehat{Y})$.

**Proposition 2.1.** $\mathrm{I}(X; \widehat{Y}) = \mathrm{I}(F(X); Y)$ *is well estimated by its empirical version (Monte-carlo approximation) with high probability, which shares the same unique (global) minimum with* $\sum_{k=1}^{n} |w^T F(x_k)|$ *at* $w^T F(x_k) = 0$, *for all* $k \in \{1, ..., n\}$.

The mutual information $\mathrm{I}(X; \widehat{Y})$ is given as

$$\mathrm{I}(X; \widehat{Y}) = \int_{\mathcal{X}} \sum_{\widehat{y} \in \mathcal{C}} p(x, \widehat{y}) \log \left( \frac{p(x, \widehat{y})}{p(x)p(\widehat{y})} \right) dx . \tag{8}$$

Apply the assumption **(II)**, the marginal distribution of $\widehat{Y}$ is uniformly distributed:

$$\frac{p(x, \widehat{y})}{p(x)p(\widehat{y})} = \frac{p(\widehat{y}|x)}{p(\widehat{y})} = 2p(\widehat{y}|x) . \tag{9}$$

Substituting (9) into (8) yields

$$\mathrm{I}(X; \widehat{Y}) = \int_{\mathcal{X}} p(x) \sum_{\widehat{y} \in \mathcal{C}} p(\widehat{y}|x) \log \left( \frac{p(x, \widehat{y})}{p(x)p(\widehat{y})} \right) dx$$

$$= \int_{\mathcal{X}} p(x) \sum_{\widehat{y} \in \mathcal{C}} p(\widehat{y}|x) \log(2p(\widehat{y}|x)) dx . \tag{10}$$

According to the Hoeffding's inequality for bounded random variables [Proposition 2.2.6, Vershynin (2018)], let $M, m$ denote upper and lower bounds of the integrand of (10) correspondingly, we have

$$\mathbb{P} \left\{ \left| \sum_{k=1}^{n} \left( \sum_{\widehat{y} \in \mathcal{C}} p(\widehat{y}|x_k) \log(2p(\widehat{y}|x_k)) - \mathrm{I}(X; \widehat{Y}) \right) \right| \geq t \right\} \leq 2e^{-\frac{2t^2}{n(M-m)^2}} . \tag{11}$$

Equivalently, with probability at least $1 - \delta$,

$$\left| \frac{\sum_{k=1}^{n} \sum_{\widehat{y} \in \mathcal{C}} p(\widehat{y}|x_k) \log(2p(\widehat{y}|x_k))}{n} - \mathrm{I}(X; \widehat{Y}) \right| < \sqrt{-\frac{\log(\frac{\delta}{2})(M-m)^2}{2n}} . \tag{12}$$

Here $\sum_{k=1}^{n} \sum_{\widehat{y} \in \mathcal{C}} p(\widehat{y}|x_k) \log(2p(\widehat{y}|x_k))/n$ is a Monte carlo estimation of RHS of $\mathrm{I}(X; \widehat{Y})$.

Recall that, for the binary case $p(\widehat{y}|x) = p(\widehat{y}|F(x))$ can expressed as

$$p(\widehat{y} = 1|F(x)) = \sigma(w^T F(x))$$
$$p(\widehat{y} = -1|F(x)) = 1 - \sigma(w^T F(x)) . \tag{13}$$

Then we have

$$\sum_{\widehat{y} \in \mathcal{C}} p(\widehat{y}|F(x_k)) \log(2p(\widehat{y}|F(x_k))) = \sigma(w^T F(x)) \log(2\sigma(w^T F(x)))$$

$$+ (1 - \sigma(w^T F(x))) \log(2 - 2\sigma(w^T F(x))) , \tag{14}$$

which is bounded by $[0, \log(2)]$.

Take $M = \log(2), m = 0$, we have

$$\mathrm{I}(X; \widehat{Y}) = \frac{\sum_{k=1}^{n} \sum_{\widehat{y} \in \mathcal{C}} p(\widehat{y}|x_k) \log(2p(\widehat{y}|x_k))}{n} + \mathrm{O} \left( \sqrt{-\frac{\log(\frac{\delta}{2}) \log(2)^2}{2n}} \right) . \tag{15}$$

hold with probability at least $1 - \delta$.

The conclusion follows from the fact that $\sum_{k=1}^{n} \sum_{\widehat{y} \in \mathcal{C}} p(\widehat{y}|x_k) \log(2p(\widehat{y}|x_k))/n$ has a unique global minimum at $w^T F(x_k) = 0$ for each $x_k$.

# B  PROOF OF PROPOSITION 2.2

Consider the training samples $\{(x_k, y_k)\}_{k=1:n}$, each $x_k$ is fed into a deep probabilistic model which outputs probability densities and predicts $\widehat{y}_k$, a realization of the prediction $\widehat{Y}$. Let $\mathcal{C} = \{\pm 1\}$ be the binary class and $n_y$ be the counts of observed occurrences of $k$ satisfying $y_k = y \in \mathcal{C}$, then $n = \sum_{y \in \mathcal{C}} n_y = \sum_y n_y$, where we omit the range over $\mathcal{C}$ for convenience.

We denote the true joint probability with $\pi_{y\widehat{y}} = p(y, \widehat{y})$, the marginal probabilities with $\pi_{y+} = \sum_{\widehat{y}} \pi_{y\widehat{y}}$ and $\pi_{+\widehat{y}} = \sum_y \pi_{y\widehat{y}}$. The mutual information $I(Y; \widehat{Y})$ can be expressed as

$$I(Y; \widehat{Y}) = I(\pi) = \sum_{y\widehat{y}} \pi_{y\widehat{y}} \log \left( \frac{\pi_{y\widehat{y}}}{\pi_{y+} \pi_{+\widehat{y}}} \right) . \tag{16}$$

Our empirical mutual information $I(\widehat{\pi})$ is defined as

$$I(\widehat{\pi}) = \sum_{y\widehat{y}} \widehat{\pi}_{y\widehat{y}} \log \left( \frac{\widehat{\pi}_{y\widehat{y}}}{\widehat{\pi}_{y+} \widehat{\pi}_{+\widehat{y}}} \right) , \tag{17}$$

where $\widehat{\pi}_{y\widehat{y}} = \frac{1}{2n_y} \sum_{i=1}^{n_y} \sigma(\widehat{y} w^T F(x_i))$.

Proposition 2.2 establishes a connection between $I(Y; \widehat{Y})$ and the cross-entropy objective for the binary case. Intuitively, increasing the confidence of the model on its correct predictions will establish a more deterministic relationship between $Y$ and $\widehat{Y}$ and thus increase the mutual information $I(Y; \widehat{Y})$.

**Proposition 2.2.** $I(Y; \widehat{Y})$ *is well estimated by* $I(\widehat{\pi})$ *with high probability, which shares the same unique (global) maximum with* $\sum_{k=1}^n \log \sigma(y_k w^T F(x_k))$ *at* $y_k w^T F(x_k) \to \infty$ *given that* $y_k w^T F(x_k) > \frac{1}{2}$*, for all* $k \in \{1, ..., n\}$.

Proposition 2.2 follows from Proposition B.1 and Proposition B.2, where Proposition B.1 shows that $I(Y; \widehat{Y})$ is well approximated by $I(\widehat{\pi})$ with high probability and Proposition B.2 shows the remaining claims in Proposition 2.2.

As shown in Lemma B.1, $\widehat{\pi}_{y\widehat{y}} = \frac{1}{2n_y} \sum_{i=1}^{n_y} \sigma(\widehat{y} w^T F(x_i))$ is an unbiased estimate of $\pi_{y\widehat{y}}$. Here $\sigma$ to represent the sigmoid function defined by $\sigma(x) = e^x / (e^x + 1)$. By leveraging the concentration property of bounded variables, *i.e.*, $\sigma(\widehat{y} w^T F(x_i))$, the estimation error can be bounded with high probability (Lemma B.2).

**Lemma B.1.** *The empirical joint probability, defined as*

$$\widehat{\pi}_{y\widehat{y}} = \frac{1}{2n_y} \sum_{i=1}^{n_y} \sigma(\widehat{y} w^T F(x_i)) , \tag{18}$$

*is an unbiased estimate of the true joint distribution* $\pi_{y\widehat{y}}$.

**Lemma B.2.** *With probability at least* $1 - \delta$*, we have*

$$|\Delta_{y\widehat{y}}| := |\widehat{\pi}_{y\widehat{y}} - \pi_{y\widehat{y}}| \le \frac{1}{2} \sqrt{\frac{\log(\frac{2}{\delta}) \min\{\frac{1}{4} \sup_x (w^T F(x))^2, 1\}}{2n_y}} . \tag{19}$$

**Proposition B.1.** *With probability at least* $1 - \delta$,

$$I(Y; \widehat{Y}) = I(\pi) = I(\widehat{\pi}) - O \left( \sqrt{\frac{\log(\frac{8}{\delta}) \min\{\frac{1}{4} \sup_x (w^T F(x))^2, 1\}}{n}} \right) . \tag{20}$$

*Proof.* To estimate the empirical mutual information given fixed samples, we use the approach by Hutter & Zaffalon (2005). In particular, taylor expansion gives

$$I(\widehat{\pi}) = I(\pi) + \sum_{y\widehat{y}} \log \left( \frac{\pi_{y\widehat{y}}}{\pi_{y+} \pi_{+\widehat{y}}} \right) \Delta_{y\widehat{y}} + O(\Delta^2) , \tag{21}$$

where $\Delta_{y\widehat{y}} = \widehat{\pi}_{y\widehat{y}} - \pi_{y\widehat{y}}$. Hence, Eq (21) together with Lemma B.2 yield, with probability exceeding $1 - |\mathcal{C}|^2\delta$,

$$\mathrm{I}(\pi) = \mathrm{I}(\widehat{\pi}) - \mathrm{O}\left(\sqrt{\frac{\log(\frac{2}{\delta})\min\{\frac{1}{4}\sup_x(w^T F(x))^2, 1\}}{n}}\right) . \tag{22}$$

Notice that, in the binary case the cardinality $|\mathcal{C}| = 2$.

$\square$

Next we prove the intermediate results, Lemmas B.1 and B.2.

*Proof of Lemma B.1.* Direct derivation on the true joint distribution $\pi_{y\widehat{y}}$ gives

$$\pi_{y\widehat{y}} = p(y, \widehat{y}) = \int_{\mathcal{X}} p(y, \widehat{y}, x)dx = \int_{\mathcal{X}} p(y|\widehat{y}, x)p(\widehat{y}|x)p(x)dx$$

$$= \int_{\mathcal{X}} p(y|x)p(\widehat{y}|x)p(x)dx$$

$$= \int_{\mathcal{X}} p(x|y)\sigma(\widehat{y}w^T F(x))p(y)dx . \tag{23}$$

Given assumption **(I)** which states that the marginal density of $Y$ is uniform over $\mathcal{C}$, for every true label $y \in \mathcal{C}$ we have $p(y) = \frac{1}{2}$.

We can therefore rewrite (23) as

$$\pi_{y\widehat{y}} = \frac{1}{2}\int_{\mathcal{X}} \sigma(\widehat{y}w^T F(x))p(x|y)dx . \tag{24}$$

According to assumption **(II)**, $p(x|y)$ is a probability density over space of signal $x$ with true label $y$.

The Monte Carlo estimation of (24) gives the empirical joint probability which is unbiased:

$$\widehat{\pi}_{y\widehat{y}} = \frac{1}{2n_y}\sum_{i=1}^{n_y} \sigma(\widehat{y}w^T F(x_i)) . \tag{25}$$

$\square$

*Proof of Lemma B.2.* Again, by leveraging the Hoeffding's inequality for bounded random variables [Proposition 2.2.6 of Vershynin (2018)], we have

$$\mathbb{P}\left\{\left|\sum_{k_y=1}^{n_y} \left(\sigma(\widehat{y}w^T F(x_{k_y})) - \mathbb{E}_{X_y}\left[\sigma(\widehat{y}w^T F(X_y))\right]\right)\right| \geq t\right\} \leq 2e^{-\frac{2t^2}{n_y(M-m)^2}} , \tag{26}$$

where $X_y$ is the data random variable whose true label is $y$.

Equivalently, with probability at least $1 - \delta$,

$$\left|\frac{\sum_{k_y=1}^{n_y} \sigma(\widehat{y}w^T F(x_{k_y}))}{n_y} - \mathbb{E}_{X_y}\left[\sigma(\widehat{y}w^T F(X_y))\right]\right| < \sqrt{\frac{\log(\frac{2}{\delta})(M-m)^2}{2n_y}} , \tag{27}$$

where $M, m$ are upper and lower bounds of random variable $\sigma(\widehat{y}w^T F(X_y))$, respectively.

Substitute (24) and (25) into (27), with probability at least $1 - \delta$,

$$|\widehat{\pi}_{y\widehat{y}} - \pi_{y\widehat{y}}| \leq \frac{1}{2}\sqrt{\frac{\log(\frac{2}{\delta})(M-m)^2}{2n_y}} . \tag{28}$$

To estimate the upper and lower bounds $M, m$ for $\sigma(\widehat{y}w^T F(X))$, we use the Taylor's theorem:

$$\sup_x \sigma(\widehat{y}w^T F(x)) = \sigma(0) + \sup_x \sigma'(c)(\widehat{y}w^T F(x))$$

$$\leq \frac{1}{2} + \frac{1}{4}\sup_x |w^T F(x)| \tag{29}$$

and

$$\inf_x \sigma(\widehat{y}w^T F(x)) = \sigma(0) + \inf_x \sigma'(c)(\widehat{y}w^T F(x))$$

$$\geq \frac{1}{2} - \frac{1}{4}\sup_x |w^T F(x)| , \tag{30}$$

given that the derivative of sigmoid function is bounded by $\frac{1}{4}$.

It follows that

$$|M - m| = \sup_x \sigma(\widehat{y}w^T F(x)) - \inf_x \sigma(\widehat{y}w^T F(x))$$

$$\leq \frac{1}{2}\sup_x |w^T F(x)| . \tag{31}$$

Also notice that $M, m$ are the bounds for sigmoid function, so their difference is at most 1.

From derivations above, we can rewrite (28) as

$$|\widehat{\pi}_{y\widehat{y}} - \pi_{y\widehat{y}}| \leq \frac{1}{2}\sqrt{\frac{\log(\frac{2}{\delta})\min\{\frac{1}{4}\sup_x (w^T F(x))^2, 1\}}{2n_y}} , \tag{32}$$

and the lemma follows. □

**Proposition B.2.** *The empirical mutual information* $\mathrm{I}(\widehat{\pi})$ *shares the same unique (global) maximum with* $\sum_{k=1}^n \log \sigma(y_k w^T F(x_k))$ *as* $y_k w^T F(x_k) \to \infty$ *given that* $y_k w^T F(x_k) > \frac{1}{2}$, *for all* $k \in \{1, ..., n\}$.

*Proof.* The empirical information $\mathrm{I}(\widehat{\pi})$ is defined by

$$\mathrm{I}(\widehat{\pi}) = \sum_{ij} \widehat{\pi}_{y\widehat{y}} \log\left(\frac{\widehat{\pi}_{y\widehat{y}}}{\widehat{\pi}_{y+}\widehat{\pi}_{+\widehat{y}}}\right) , \tag{33}$$

where the empirical joint probability is given by

$$\widehat{\pi}_{y\widehat{y}} = \frac{1}{2n_y}\sum_{i=1}^{n_y} \sigma(\widehat{y}w^T F(x_i)) . \tag{34}$$

It then follows that for any $y \in \mathcal{C}$,

$$\widehat{\pi}_{y+} = \frac{1}{2n_y}\sum_{i=1}^{n_y} \sigma(w^T F(X_i)) + \frac{1}{2n_y}\sum_{i=1}^{n_y} \sigma(-w^T F(X_i))$$

$$= \frac{1}{2n_y}\sum_{i=1}^{n_y} (\sigma(w^T F(X_i)) + \sigma(-w^T F(X_i)))$$

$$= \frac{1}{2n_y}\sum_{i=1}^{n_y} 1 = \frac{1}{2} . \tag{35}$$

In binary case it means that

$$\widehat{\pi}_{1(-1)} = \frac{1}{2} - \widehat{\pi}_{11} , \quad \widehat{\pi}_{(-1)1} = \frac{1}{2} - \widehat{\pi}_{(-1)(-1)} \tag{36}$$

and the empirical mutual information can decomposed as

$$
\mathrm{I}(\widehat{\pi}) = \widehat{\pi}_{11} \log\left(\frac{\widehat{\pi}_{11}}{\widehat{\pi}_{11} + \frac{1}{2} - \widehat{\pi}_{(-1)(-1)}}\right) + \left(\frac{1}{2} - \widehat{\pi}_{11}\right) \log\left(\frac{\frac{1}{2} - \widehat{\pi}_{11}}{\frac{1}{2} - \widehat{\pi}_{11} + \widehat{\pi}_{(-1)(-1)}}\right) +
$$
$$
\left(\frac{1}{2} - \widehat{\pi}_{(-1)(-1)}\right) \log\left(\frac{\frac{1}{2} - \widehat{\pi}_{(-1)(-1)}}{\widehat{\pi}_{11} + \frac{1}{2} - \widehat{\pi}_{(-1)(-1)}}\right) + \widehat{\pi}_{(-1)(-1)} \log\left(\frac{\widehat{\pi}_{(-1)(-1)}}{\frac{1}{2} - \widehat{\pi}_{11} + \widehat{\pi}_{(-1)(-1)}}\right) + \log(2) \ .
$$
$$(37)$$

We differentiate (37) with respect to $\widehat{\pi}_{11}$ and $\widehat{\pi}_{(-1)(-1)}$, and calculate the critical points over the domain $[0, \frac{1}{2}]$ for both variables, which gives

$$
\widehat{\pi}_{11} = \widehat{\pi}_{(-1)(-1)} = \frac{1}{4} \ . \tag{38}
$$

Observe that (38) is a global minimum over $[0, \frac{1}{2}] \times [0, \frac{1}{2}]$. Since this is the unique critical point where the derivative vanishes, the global maximums can only be obtained on the boundaries. In particular, if we restrict $(\widehat{\pi}_{11}, \widehat{\pi}_{(-1)(-1)})$ on $[\frac{1}{4}, \frac{1}{2}] \times [\frac{1}{4}, \frac{1}{2}]$, (37) is a strictly increasing function over $\widehat{\pi}_{11}, \widehat{\pi}_{(-1)(-1)}$ and the unique global maximum is obtained at

$$
\widehat{\pi}_{11} = \widehat{\pi}_{(-1)(-1)} = \frac{1}{2} \ . \tag{39}
$$

The proposition follows from the definition (34) of $\widehat{\pi}_{y\widehat{y}}$ that (39) is only approached when $y_k w^T F(x_k) \to \infty$, for all $k \in \{1, ..., n\}$.

$\square$

## C  INVERTIBILITY IS BENEFICIAL

We show in Proposition C.1 that the lower bound for the classification error is itself lower bounded by a constant, which is attained if $F$ is invertible. Although the performance of the model also depends on the classifier $w$, our bound claims that an invertible feature map $F$ could provide a better environment for the classifier $w$ to perform well. Intuitively, invertibility preserves the information of the signal $X$ as it flows through the neural network and reaches the classifier $w$; on the other hand, $w$ potentially performs better on the input that preserves all information of the data compared to the one that doesn't.

**Proposition C.1.** *(Fano's Inequality) The classification error is lower bounded as follows:*

$$
\mathbb{P}(Y \neq \widehat{Y}) \geq \frac{H(Y|F(X)) - \log(2)}{\log(|\mathcal{C}| - 1)} \ . \tag{40}
$$

*The lower bound satisfies*

$$
\frac{H(Y|F(X)) - \log(2)}{\log(|\mathcal{C}| - 1)} \geq \frac{H(Y|X) - \log(2)}{\log(|\mathcal{C}| - 1)} \tag{41}
$$

*for all $F$, and the equality is attained if $F$ is invertible.*

Let $Z = F(X)$ and the machinery of deep learning can be decribed by the following Markov Chain:

$$
Y \to X \to Z \to \widehat{Y} \ . \tag{42}
$$

Lemma C.1 is a technical result that helps to prove Proposition C.1. The information $Z = F(X)$ has about the true labels $Y$ is maximized when $F$ is invertible, which is beneficial in the sense that the key information influential for classification can be well preserved.

**Lemma C.1** (Chain Rule). *Given the Markov Chain assumption equation 42, we have*

$$
\mathrm{I}(Y; \widehat{Y}) \leq \mathrm{I}(Y; Z) \leq \mathrm{I}(Y; X) \ , \tag{43}
$$

*and the second equality is attained if $F$ is invertible.*

*Proof.* We will only prove the second inequality and the first inequality follows by a similar argument. Consider the decomposition

$$
\begin{aligned}
\mathrm{I}(Y;X,Z) &= \int_{\mathcal{X}} \sum_y \int_{\mathcal{Z}} p(x,y,z) \log \frac{p(x,y,z)}{p(y)p(x,z)} dx dz \\
&= \int_{\mathcal{X}} \sum_y \int_{\mathcal{Z}} p(x|y,z)p(y,z) \log \frac{p(x|y,z)p(y,z)}{p(y)p(x|z)p(z)} dx dz \\
&= \mathrm{I}(Y;Z) + \int_{\mathcal{X}} \sum_y \int_{\mathcal{Z}} p(x|y,z)p(y,z) \log \frac{p(x|y,z)}{p(x|z)} dx dz \\
&= \mathrm{I}(Y;Z) + \int_{\mathcal{X}} \sum_y \int_{\mathcal{Z}} p(x,y|z)p(z) \log \frac{p(x,y|z)}{p(x|z)p(y|z)} dx dz \\
&= \mathrm{I}(Y;Z) + \mathrm{I}(X;Y|Z) .
\end{aligned}
\tag{44}
$$

Similarly we obtain

$$
\mathrm{I}(Y;X,Z) = \mathrm{I}(X;Y) + \mathrm{I}(Y;Z|X) .
\tag{45}
$$

equation 44 together with equation 45 yields

$$
\mathrm{I}(Y;Z) + \mathrm{I}(X;Y|Z) = \mathrm{I}(X;Y) + \mathrm{I}(Y;Z|X) .
\tag{46}
$$

According to the Markov Chain setting, $Y$ and $Z$ are conditionally independent given $X$, hence $\mathrm{I}(Y;Z|X) = 0$; in addition, the mutual information $\mathrm{I}(X;Y|Z)$ is nonnegative. It follows from (46) that

$$
\mathrm{I}(Y;Z) \le \mathrm{I}(Y;X) .
\tag{47}
$$

□

Next we present a lower bound for the classification error. This lower bound is negatively related to the mutual information $I(Y;F(X))$, and it attains its minimum if $F$ is invertible. Although the performance also depends on the classifier $w$, Proposition C.1 implies that an invertible feature map $F$ allows more chances for the classifier $w$ to perform well.

*Proof of Proposition C.1.* Consider the random variable $E$ defined as:

$$
E = \begin{cases} 1, & \text{if } Y \ne \widehat{Y} \\ 0, & \text{otherwise .} \end{cases}
\tag{48}
$$

By the Chain Rule following from similar arguments presented in Lemma C.1, we have

$$
\begin{aligned}
H(E,Y|\widehat{Y}) &= H(Y|\widehat{Y}) + H(E|Y,\widehat{Y}) \\
H(E,Y|\widehat{Y}) &= H(E|\widehat{Y}) + H(Y|E,\widehat{Y}) .
\end{aligned}
\tag{49}
$$

Note that $H(E|Y,\widehat{Y}) = 0$, since the value of $E$ is determined given the knowledge of $Y, \widehat{Y}$. It then follows that

$$
\begin{aligned}
H(Y|\widehat{Y}) &= H(E|\widehat{Y}) + H(Y|E,\widehat{Y}) \\
&\le \log(2) + H(Y|E=0,\widehat{Y})\mathbb{P}(E=0) + H(Y|E=1,\widehat{Y})\mathbb{P}(E=1) \\
&= \log(2) + H(Y|E=1,\widehat{Y})\mathbb{P}(Y \ne \widehat{Y}) \\
&\le \log(2) + \log(|\mathcal{C}|-1)\mathbb{P}(Y \ne \widehat{Y}) .
\end{aligned}
\tag{50}
$$

On the other hand, Lemma C.1 shows that

$$
H(Y) - H(Y|\widehat{Y}) = \mathrm{I}(Y;\widehat{Y}) \le \mathrm{I}(Z;Y) = H(Y) - H(Y|\widehat{Z}) ,
\tag{51}
$$

which gives

$$H(Y|Z) \leq H(Y|\widehat{Y}) \ . \tag{52}$$

Substitute it into (50) yields the result

$$H(Y|Z) \leq \log(2) + \log(|\mathcal{C}| - 1)\mathbb{P}(Y \neq \widehat{Y}) \ . \tag{53}$$

As for the second statement, Lemma C.1 shows that

$$H(Y) - H(Y|Z) = \mathrm{I}(Y; Z) \leq \mathrm{I}(Y; X) = H(Y) - H(Y|X) \ . \tag{54}$$

It then follows that,

$$H(Y|Z) = H(Y|F(X)) \geq H(Y|X) = 0 \ , \tag{55}$$

where the equality is attained if $F$ is invertible. $\qquad\square$

## D  INVERTIBILITY OF RESNET

ResNet is designed to allow the model to "learn" the identity map easily. Specifically, the input vector $x$ and output vector $y$ of a building block are related by

$$y = \mathcal{L}(x) + x = (\mathcal{L} + I)(x) \ , \tag{56}$$

where the operator $\mathcal{L}$ could be a composition of activation functions, convolutions, drop-out(Srivastava et al. (2014)) and batch normalization(Ioffe & Szegedy (2015)). It's shown in Lemma D.1 below that if the operator norm $|\mathcal{L}| < 1$, then $\mathcal{L} + I$ is theoretically guaranteed to have an inverse, which enables information preservation among intermediate layers. In Figure 5 we experimentally verify that $|\mathcal{L}| < 1$ for all building blocks during the training process. In general, operations such as ReLU, pooling, drop-out are not invertible(Dosovitskiy & Brox, 2015); it's challenging to build a strictly invertible network (Jacobsen et al., 2018). From this point of view, the beauty of ResNet lies in the fact that it's guaranteed to be invertible regardless how $\mathcal{L}$ evolves during the training process, as long as $|\mathcal{L}| < 1$.

Although the usual design of ResNet does involve non-invertible components such as pooling, we argue that ResNet still has descent invertible property compared to the majority of other neural network designs. We also experimentally verify that our regularization does not improve a very deep ResNet on its performance by a clear margin; we speculate that information will lose more as it goes deeper.

**Lemma D.1.** *Consider Holder Space $C^{0,1}(\overline{U})$, where $\overline{U}$ is the closure of some bounded open set $U$, with equiped Holder norm $|\mathcal{L}| = \alpha \sup_{x \in U} |\mathcal{L}(x)| + \sup_{x,y \in U, x \neq y} \left\{ \frac{|\mathcal{L}(x) - \mathcal{L}(y)|}{|x - y|} \right\}$, here $\alpha$ is some positive scalar. If $\mathcal{L} \in C^{0,1}(\overline{U})$ and $|\mathcal{L}| < 1$, then there exists $\mathcal{B}$ such that $\mathcal{B}(I + \mathcal{L}) = (I + \mathcal{L})\mathcal{B} = I$.*

*Proof.* It's well known that $C^{0,1}(\overline{U})$ is a Banach space (Lax).

Define

$$\mathcal{B} = \sum_{n=0}^{\infty} (-\mathcal{L})^n \ . \tag{57}$$

Since $|\mathcal{L}| < 1$, (57) is a Cauchy sequence and coverges in Banach space. Convergence sequences can be multiplied termwise, it follows that

$$\mathcal{B}\mathcal{L} = \mathcal{L} \sum_{n=0}^{\infty} (-\mathcal{L})^n = -\sum_{n=1}^{\infty} (-\mathcal{L})^n = -(\mathcal{B} - I) \ . \tag{58}$$

So $\mathcal{B}(I + \mathcal{L}) = I$. The other equality $(I + \mathcal{L})\mathcal{B} = I$ can be shown to hold in the same fashion. $\qquad\square$

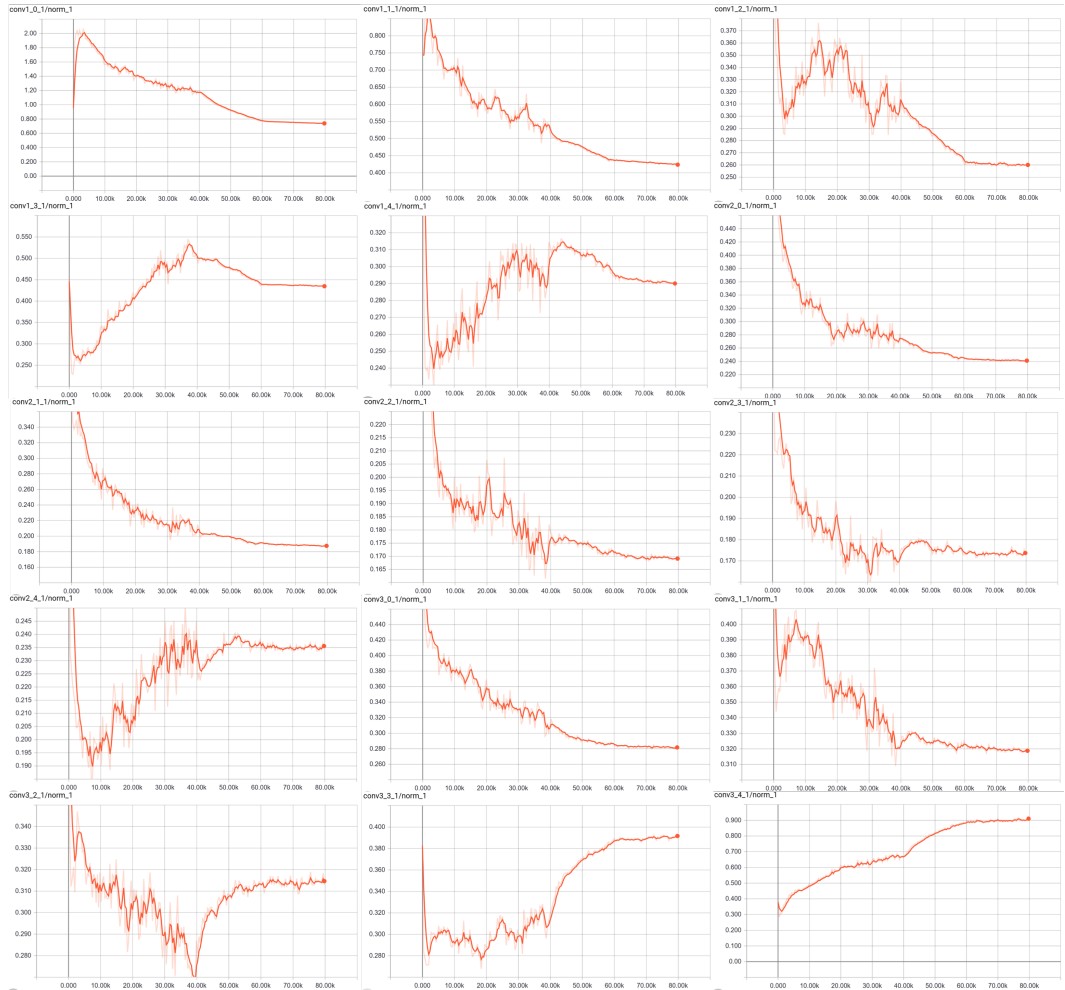

Figure 5: We measure the operator norm of $\mathcal{L}$ in each building block of ResNet-32 over 80k training steps. It can be observed that, the operator norms are all bounded by 1 throughout the training process, which verifies the hypothesis made in Appendix D. We conlude that ResNet is invertible.

## E  IMPLEMENTATION DETAILS

We implement all models using Tensorflow. We modify the ResNet based on the code provided at `https://github.com/tensorflow/models/tree/master/research/resnet`. We follow the same learning rate scheme proposed in the original code. For the InvNet on MNIST, we train the network with initial learning rate 0.1, and decay it by 0.7 every 10 epochs. For both InvNet and ResNet, we apply the $\ell_1$ norm regularization every 30 iterations.

# F    RESULTS FOR ALL DIGITS ON INVNET

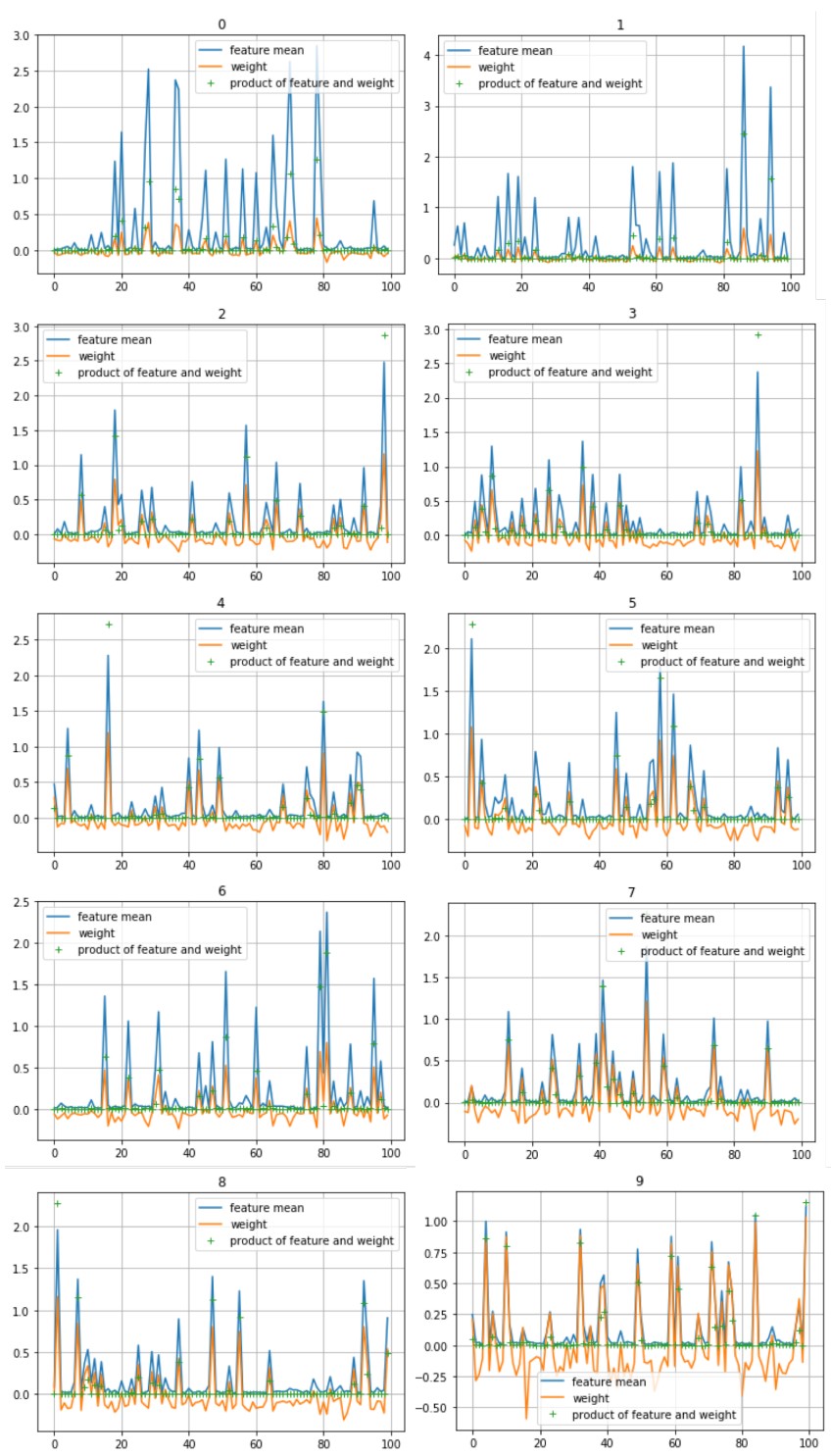

Figure 6: A reproduction of the feature statistical results of InvNet on MNIST for all digits

This is a reproduction of the feature statistical results of InvNet on MNIST for all digits for Figure 2.

We observe that each digits have their specific entries with high value assigned to both weights and feature means. An additional observation is that typically the feature entry with low mean also has low standard deviation, such entry rarely contributes to the logits for classification if the corresponding value of weight is also small.

# G   RESULTS ON I-REVNET

We reproduce the results in Section 4.2 on i-RevNet (Jacobsen et al. (2018)). The statistics and ROC curves of the features produced by i-RevNet and Reg-i-RevNet on CIFAR10 is similar to those presented in Section 4.2.

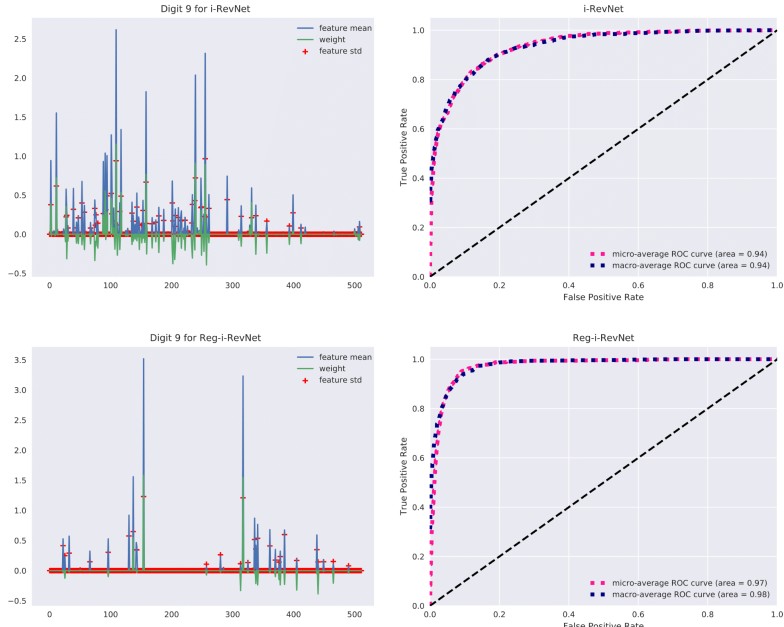

Figure 7: Reproduction of Experimental Results in Section 4.2 on i-RevNet. The plots in the first row are for i-RevNet and the plots in the second row are for Reg-i-RevNet.

# H   HISTOGRAM ON FEATURES OF DIGIT 9

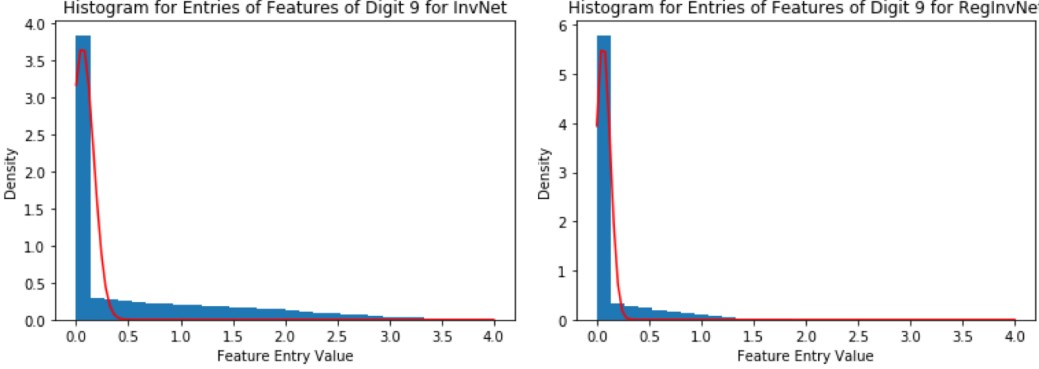

Figure 8: The histogram for values of feature entries of Digit 9

The histogram for values of feature entries of Digit 9 has a decaying shape but with a heavier tail compared to that of Gaussian with small variance. The spasity of w depends on our choice of hyperparameters. For example in Figure 4 we measure the sparsity of the learned w of RegResNet for CIFAR10: about $60\%$ entries of w are precisely zero.

## I THE USE OF SURROGATE FUNCTIONS TO MINIMIZE MUTUAL INFORMATION

Our objective is

$$I(X, \widehat{Y}) = H(\widehat{Y}) - H(\widehat{Y}|X) . \tag{59}$$

Assume $\widehat{Y}$ is uniformly distributed on $\mathcal{C}$, $H(\widehat{Y})$ becomes a constant and we have

$$
\begin{aligned}
-H(\widehat{Y}|X) &= -\int_{\mathcal{X}} \sum_{\widehat{y} \in \mathcal{C}} p(x, \widehat{y}) \log \frac{p(x, \widehat{y})}{p(x)} \\
&= -\int_{\mathcal{X}} p(x) \sum_{\widehat{y} \in \mathcal{C}} p(\widehat{y}|x) \log p(\widehat{y}|x) \\
&\approx -\sum_{x \in \text{Bat}} \sum_{\widehat{y} \in \mathcal{C}} p(\widehat{y}|x) \log p(\widehat{y}|x)
\end{aligned} \tag{60}
$$

Note that (60) is composed of functions in the form $a \log a$ where $a$ is the output of a softmax function on logits. For simplicity we consider the binary case where $a = \left(1 + e^{-w^T F(x)}\right)^{-1}$. The derivative of it with respect to $w$ takes the form

$$\frac{e^{-w^T F(x)} F(x)}{\left(1 + e^{-w^T F(x)}\right)^2} . \tag{61}$$

Assume $|w^T F(x)|$ is large, if $w^T F(x) > 0$ then the numerator decays exponentially with respect to $|w^T F(x)|$ and the denominator converges to 1; if $wF(x) < 0$ then the denominator grows exponentially with respect to $|w^T F(x)|$ and dominates the numerator. To conclude, for large $|w^T F(x)|$ the gradient is decaying to zero exponentially. So the effect of punishing large logits $|w^T F(x)|$ by this objective is not clear as the gradient vanishes for large $|w^T F(x)|$. We also analyzed the general softmax functions for multi-labels and found they exhibit similar properties.

We propose to use a surrogate function $w^T F(X)$ to minimize this objective to make the regularization effect more clear. We prove that our regularizer achieves the same goal compared to the mutual information objective but provides a better gradient for training.

