# OpenReview forum: "Information Regularized Neural Networks"
_ICLR.cc/2019/Conference_

### Official Review · AnonReviewer2 · 2018-10-30
**Review (updated after readng other reviews and other responses)**

**Rating:** 5
**Confidence:** 3

**Review:**

In this paper, the authors propose to train a model from the point of view of maximizing mutual information between the predictions and the true outputs, with a regularization term that minimizes irrelevant information while learning. They show that the objective can be minimized by looking to make the final layer vectors be as uncorrelated as possible to the final layer representations, and simplify the same by applying Holder’s inequality to make the optimization tractable. They also apply an L1 penalty on the final layer. Experiments on CIFAR and MNIST show that using their regularizer to train DNN models yield  gains in performance. The presence of the L1 penalty also makes the results more interpretable (to the extend possible by looking at a subset of features in the last layer of a DNN).

COMMENTS:

- Meta Point: To really see that the regularization framework you’re proposing is good, why not just pick a simple, feedforward model or convNet, see the performance and then compare it with the regularizer you’re proposing? That will help hit the point home.

- Page 1: before jumping to equations (1) and (2), please formally define Mutual Information. The actual definition is much later in the text, but it’s better to define first.

- Beyond  referring the user to section 3 on Page 1, please also mention a couple of key references in the appropriate locations.

- Page 3 paragraph 2: “Mutual information is bounded … correct them” : Can you provide some formulas for this and make this concrete? Or perhaps provide some references? This line is vague.

 - prop 2.1 and 2.2: can you define what you mean by “empirical version”? Again, it’s probably good to have these terms crisply defined before using them.

- eqn (6) is interesting. Holder’s inequality gives you the product terms. Then you can also apply the AM-GM inequality, and get a sum. So then at the end of it all, you’re left with the standard elastic net penalty and not the product form. In that case, aren’t we back to just the usual regularization strategy? And in which case, should I interpret the results you have in sec 4 as “using L1 penalties with L2 is good” ?

- To the point above, I guess one difference after the AM-GM step is that you will not have a squared L2 norm, but just L2. This is reminiscent of linear models where they use L2 loss instead of squared L2 loss. But on the penalty, squaring just adds smoothness. Can you comment on this?

- sec 4.2.1: I don’t see how fig (2) (L) is “roughly Gaussian”. Can you explain? Maybe plot the histogram? Also for fig (2, R): the coefficients are approximately sparse. It’s not sparse as you claim since there are almost no zeros in the coefficients.

- I don’t get the point of sec 4.3: How does this claim not apply to all deep learning models, regardless of the penalizations you propose?


edit:
I have read the responses and the other reviews. The authors have addressed the few major points I had. I still think there are a few gaps that need to be addressed (as pointed by the other reviewers)

---

> ### Author Response · Authors · 2018-11-22
> **Response to Reviewer 2**
>
> We thank for the reviewer for the thoughts and comments!  We provide detailed response to each comment as below.
>
> 1. "comment on AM-GM inequality for L2 loss"
>
> The naive l2 regularization only involves regularizing the parameters w. Our derivation on the mutual information objective shows that feature regularization on F(X) is also necessary for good generalization of deep neural network. We observe that naive regularization on w results in a blow up on the norm of F(X), meaning the neural network can absorb the regularization effect by upscaling the feature. We show in Figure 4 that our regularization can hold the magnitude of F(X) while compressing w.
>
> As mentioned in (Neyshabur et al., 2015), neural networks are equivalent up to some scaling factors passed among layers. As a consequence, there exist unbalanced neural networks with large l2 weights but are equivalent to those with small l2 weights. So the common belief that l2 regularization can "simplify" a model does not necessarily make sense for deep model.
>
>
> 2. "pick a simple, feedforward model or convNet, see the performance and then compare it with the regularizer"
>
> We perform our regularization on a simple structure named "34-layer plain" in (He et al., 2015) and the result on CIFAR10 is shown in Table 2. There are some marginal improvement on this network but it's less robust against the choice of our hyperparameters.
>
> Our theoretical framework is built upon the assumption that the models have decent invertibility property. We emphasize that invertible neural networks are an important class of deep networks by citing the related work in Section 3 and proving in Appendix C that lower bound for the classification error is itself lower bounded by a constant, which is attained if the network is invertible. We perform our experiments on ResNet because we observe it's composed of blocks in a intrinsic invertible functional form I+L so we expect it has a good invertibility property. We adopt the suggestion from Reviewer 1 that we should also perform all our experimental results on more theoretically grounded invertible neural networks such as i-RevNet.
>
>
> 3. "sec 4.3: How does this claim not apply to all deep learning models, regardless of the penalizations you propose"
>
> In the literature of machine learning l2 regularization only makes sense for shallow structure like logistic regression and SVM. For logistic regression, we can interpret it with heuristics from Occam's razor or, from a probablitic point of view, a Gaussian prior. For SVM it means a larger margin for the support features. But these interpretation becomes less intuitive for deep learning as the interactions between parameters become extremely complicated. In our work we reduce the deep structure into a linear one with MI objective and invertibility of ResNet, and formally justify the use of l2 regularization on both features F(X) and parameters w in the last layer; under this setting we interpret the meaning of l1/l2 regularization for deep models from classical perspectives in Section 4.3.
>
> We fix our description in the revised version.
>
>
> 4. "I don’t see how fig (2) (L) is “roughly Gaussian”"
> "Also for fig (2, R): the coefficients are not sparse as you claim"
>
> We agree that the use of the term "roughly Gaussian" is imprecise. We have plotted the histogram of the values of feature entries of digit 9 in Appendix H of the revised version.
>
> Consider an perturbed image by some Gaussian noise, if the noise were to make an impact to the output of the model, it must modify the values of feature entries where the model assigns the corresponding weights with high values. But for our regularized model, the number of weights with high values is smaller compared to that of normal model so it's harder for a random noise to make huge impact to the output.
>
> We fix our use of terms in the revised version.
>
>
> 5. "“Mutual information is bounded … correct them”:Can you provide some formulas for this and make this concrete"
>
> We explain in detail in Appendix I of the revised version. Neural networks in known to be occasionally over-confident in its prediction which is in fact wrong. In particular we want to punish some logits with large absolute value |w^F(X)| but have the wrong "sign" (for binary case). We show that information objective does not provide good gradient to fix this problem.
>
>
> 6. "prop 2.1 and 2.2: can you define what you mean by “empirical version”"
>
> We have added the necessary definition in our proposition stated in the main text of the revised version. In general by empirical version we mean a Monte Carlo approximation of the population quantities. We show in our proof that if the sample size N is large enough, our Monte Carlo approximation is accurate with high probability.
>
>
> We have fixed the citation and definition of MI as recommended in the revised version.

---

### Official Review · AnonReviewer3 · 2018-11-01
**This paper has encouraging experimental result and the formulation is plausible, but I'm confused about how the proposed model tends to overlook irrelevant information.**

**Rating:** 6
**Confidence:** 3

**Review:**

This paper proposed to decompose the parameters into an invertible feature map F and a linear transformation w in the last layer. They aim to maximize mutual information I(Y, \hat{Y}) while constraining irrelevant information, which further transfers to regularization on F and w. The authors also spend pages explaining how the hyper-parameters can be chosen.

Comments:
1. The experimental results showed a noticeable improvement on CIFAR-100 and is fairly robust to alpha_2.
2. The formulation seems plausible.
3. For Figure 2 and discussion in Section 4.2.1, I'm less convinced that the entries with high feature mean is 'relevant' and the others are not by looking at just digit 9 samples. For example, an entry with small feature mean should still be given high w_10 value if for all other 9 digits the same entry has even smaller feature mean.

--------UPDATE AFTER READING THE AUTHORS' COMMENTS-----------
1. Appendix F lacks explanation. So I'm going to say what I meant in details.

In order to achieve high accuracy the model must assign high values on some entries of weights to separate the different classes. w_10 is a linear separator, not necessarily entry-wise (unless the features are independent).
I would take 1k features of each class and compute their principal components. Check if these components are different from class to class and plot the dot product of components and weights. If the following happens I would be more convinced:
1) principal components of digit 0 and digit 9 differs a lot AND
2) w_0 weights components of digit 0 higher but weights those of digit 9 lower

2. "But for our regularized model, the number of weights with high values is smaller compared to that of normal model ..."
I'm not convinced. When perturbed by Gaussian noise, the variance on output does not necessarily depends on sparsity. In fact, it depends on the norm of the weights.

---

> ### Author Response · Authors · 2018-11-22
> **Response to Reviewer 3**
>
> We thank the reviewer for the comments! We provide the detailed response to questions as below.
>
> 1. "an entry with small feature mean should still be given high w_10 value if for all other 9 digits the same entry has even smaller feature mean"
>
> Neural networks should assign high w_10 values to feature entries of 9 with high values, so their product can contribute to the logits significantly. We agree that neural networks may also assign high w_10 values to feature entries of 9 with small absolute value but relatively large compared with the same feature entries of other digits; in this case we expect w_10 should hold even higher value so the product w^TF(X) contributes significantly to the logits.
>
> Our derivation in (6) shows that we should regularize the inner product (w^TF(X))_i for class i, which is the sum of product from each entry. It is possible that for one entry we have small feature value but large classifier value, but if the product of them is relatively small compared to other entry product then we will not consider it as an important feature entry to classification.
>
> We have reproduced the feature statistics plot of all digits for InvNet on MNIST in Appendix F of the revised version. We observe that each digits have their specific entries with high value assigned to both weights and feature means.
>
>
> 2. "how the proposed model tends to overlook irrelevant information"
>
> Consider an image perturbed by some Gaussian noise, if the noise were to make an impact to the output of the model, it must significantly modify the values of feature entries where the model assigns the corresponding weights with high values. But for our regularized model, the number of weights with high values is smaller compared to that of normal model so it's harder for a random noise to make huge impact to the output (unless this noise is maliciously designed).
>
> Our belief is under our regularization, the weight w is shaped into a "sparse" form adapted to the particular input data distribution, so it is hard for any irrelevant information induced on the data to make an impact to the model's output.

---

> ### Author Response · Authors · 2018-11-30
> **Response to Reviewer 3 Continued(1)**
>
> 1.1. Similarity of feature spaces
>
> We calculate the principle component of 1000 features of each digit. To measure the similarity among the subspace generated by the top 5 principle components of each digit, we using the following "metric":
> let U and V be 100*5 matrices storing the principle component of features of class i and j, define the projection matrix onto the subspace as
> P_U = U*(U^T*U)^{-1}*U^T, P_V = V*(V^T*V)^{-1}*V^T
> Then if x is a vector lying in the intersection of spaces generated by U and V, x should be invariant under the projections:
> x = P_U*x, x = P_V*x
> It follows that x is an eigenvector of matrix P_U*P_V with eigenvalue precisely 1.
> In fact the eigenvalues of P_U*P_V range in [0,1].
>
> We use the SUM OF EIGENVALUES P_U*P_V to measure the similarity between subspaces generated by columns of U and V.
> Larger sum indicates more similarity.
>
> We choose not to use U^T*V as a measure because in high dimensional space, vectors tend to be orthogonal to each other so the resulting product does not give too much information.
>
> We show result as follows:
>
> InvNet
>         0  |  1  |  2  |  3   |  4  |  5  |  6  |  7  |  8  |  9   | Most Similar Digit
> 0|   *    0.16 0.93 0.50 0.36 0.85 0.81 0.54 0.66 0.68|       2,5
> 1| 0.16   *    0.64 0.36 0.68 0.33 0.37 0.53 0.55 0.47|       2,4
> 2| 0.93 0.64   *    1.15 0.64 0.31 0.77 0.91 0.87 0.54|       0,3
> 3| 0.50 0.36 1.15   *    0.31 1.44 0.35 0.85 0.93 0.62|       2,5
> 4| 0.35 0.68 0.64 0.32   *    0.47 0.83 0.84 0.78 1.66|       7,9
> 5| 0.85 0.33 0.31 1.44 0.47   *    0.71 0.68 0.94 0.80|       3,8
> 6| 0.81 0.37 0.77 0.35 0.83 0.71   *    0.29 0.45 0.31|       0,4
> 7| 0.54 0.53 0.91 0.85 0.84 0.68 0.29   *    0.71 1.02|       2,9
> 8| 0.66 0.55 0.87 0.93 0.78 0.94 0.45 0.71   *    1.11|       5,9
> 9| 0.68 0.47 0.54 0.62 1.66 0.80 0.31 1.02 1.11   *   |       4,8
> Mean:1.11, Std:1.33
>
> RegInvNet
>         0  |  1  |  2  |  3   |  4  |  5  |  6  |  7  |  8  |  9   | Most Similar Digit
> 0|   *    0.15 0.73 0.64 0.35 0.50 1.23 0.36 0.79 0.29|       6,8
> 1| 0.15   *    1.06 0.63 0.57 0.36 0.15 0.91 0.44 0.60|       2,7
> 2| 0.73 1.06   *    0.85 0.25 0.16 0.34 0.94 0.66 0.52|       1,7
> 3| 0.64 0.63 0.85   *    0.15 1.41 0.29 0.71 1.31 0.51|       5,8
> 4| 0.35 0.57 0.25 0.15   *    0.21 0.91 0.90 0.39 1.19|       6,9
> 5| 0.50 0.36 0.16 1.41 0.21   *    1.15 0.50 1.22 0.39|       3,8
> 6| 1.23 0.15 0.34 0.29 0.91 1.15   *    0.25 0.98 0.31|       0,5
> 7| 0.36 0.91 0.94 0.71 0.90 0.50 0.25   *    0.34 1.08|       2,9
> 8| 0.79 0.44 0.66 1.31 0.39 1.21 0.98 0.34   *    0.85|       3,5
> 9| 0.29 0.60 0.52 0.51 1.19 0.39 0.31 1.08 0.85   *   |       4,7
> Mean:1.07, Std:1.35
>
> Our regularization does not improve the seperation among the feature spaces of different digits as the mean and standard deviation of the results do not differ too much between InvNet and RegInvNet. But after a close inspection, we observe that the feature learned by our regularization gives information on the similarity between features that is closer to human's conception. For example, if we compare the Most Similar Digit result from RegInvNet and InvNet, we find the following that matches our intuition
> - 0 is more similar to 6,8 compared to 2,5
> - 1 is more similar to 7 compared to 4
> - 3 is more similar to 8 compared to 5
> - 6 is more similar to 5 compared to 4
> - 8 is more similar to 3 compared to 9
> - 9 is more similar to 7 compared to 8
>
> Our explaination to this fact is that the features learned by the model should encode information about the relationship among the digits with different classes but not live in completely different space away from others. This idea is similar to the motivation of distillation (Hinton et al., 2015).
>
> -------------------------------------------------------------------------------------------------------
> 1.2. Difference in predictions
>
> Although the features alone learned from our regularizer share meaningful information, when they get multiplied by our classifier w, the output is clearly distinguished.
>
> We perform the dot product between principle components and weights and find the follows:
>
> -maximum of dot product between all principle components and the normalized (i+1)th column of w for each digit i
> InvNet
>   0   |  1  |  2  |  3   |  4 |  5  |  6   |  7  |  8  |  9  |
> 0.37 0.44 0.41 0.47 0.48 0.48 0.62 0.35 0.44 0.35
>
> RegInvNet
>   0   |  1  |  2  |  3   |  4 |  5  |  6   |  7  |  8  |  9  |
> 0.93 0.95 0.90 0.92 0.88 0.93 0.95 0.94 0.91 0.93
>
> -maximum of dot product between all principle components of digit 0/9 and the normalized 10th/1st column of w
> InvNet
> 0.40 / 0.30
>
> RegInvNet
> 0.29 / 0.45
>
> We see that our classifier can find (almost) precisely the direction of important principle component (we observe it's either the first or the second) of the corresponding features. The interaction between classifier for digit0/9 and the principle components of feature of digit9/0 is similar between InvNet and RegInvNet.
>
> Our conlusion is our features are more meaningful and our classifier is sharper.

---

> ### Author Response · Authors · 2018-11-30
> **Response to Reviewer 3 Continued(2)**
>
> 2. Yes the variation on outputs also depends on the weights. We would like to argue that the variation has less effect on the inner product w^TF(X). Suppose random perturbation on the image leads to random variation \epsilon on the feature F(X). On one hand regularizing w gives smaller perturbation to the product w^T\epsilon. On the other hand since perturbation does not affect w, we can analyze the statistics of F(X)+\epsilon with perturbation. We also plot the standard deviation in Figure 2 for each feature entry of 9 and observe that feature entries remain having relatively high values when putting the deviation into consideration, which is not the case for InvNet. We conclude that due to this high contrast of mean values among feature entries, the classifier can still make the correct prediction under variations of F(X).

---

### Official Review · AnonReviewer1 · 2018-11-03
**Theoretically grounded regularizer that penalizes confident predictions, experimental section needs to be improved**

**Rating:** 6
**Confidence:** 4

**Review:**


The authors propose a regularizer placed on the final linear layer of invertible networks that penalizes confident predictions, leading to better generalization. The algorithm is theoretically grounded and even though SOTA networks do not meet some theoretical requirements in practice, it seems to be effective.

The ideas presented are interesting, but the paper is confusing at times and some motivations seem hand-wavy (see below).

Even though penalizing overly confident predictions is an important topic, it has been attacked by various approaches in the past. It is not clear how the proposed method empirically compares to other approaches from the literature. On the theoretical side, proposition 2.1 and its proof are the main contribution. This very interesting observation could potentially be very useful in many tasks and shows once again why invertible neural networks are an important class of deep networks.

Main concerns:

The authors do not compare their method to other approaches from the literature with similar goals, such as [1]. Therefore, it is hard to judge the performance of the proposed regularizer.

The authors claim that their InvNet is approximately invertible but there is no guarantee for this, making empirical conclusions unclear. The experiments would be more conclusive if a network that is fully invertible by construction is used. Such networks exist and perform on par with ResNets [2], so there is no reason not to use them. This would remove the need for analysis or discussion of this matter, as this issue clutters the main contribution and makes the claims rather fuzzy right now.

Minor

- Why are citations displayed in blue? This does not seem to be ICLR formatting standard.

[1] Pereyra et al., "Regularizing neural networks by penalizing confident output distributions."
[2] Jacobsen et al., "i-RevNet: Deep Invertible Networks"

---

> ### Author Response · Authors · 2018-11-22
> **Response to Reviewer 1**
>
> We thank the reviewer for the insightful suggestions!
>
> 1. We have implemented i-RevNet (Jacobsen et al., 2018) with our regularization, CP - confidence penalizing with entropy (Pereyra et al., 2017) and LS - label smoothing (Szegedy et al., 2015) over some choices of hyperparameters. The performance results on CIFAR100 over 5 trails are provided below. We also reproduce other experiments on i-RevNet in Appendix G of the revised version.
>
> ------------------------------------------------------------------------------------------------------------------------------------------------------------
>             | baseline |                         Our Regularization                               |                CP                  |                 LS               |
> ------------------------------------------------------------------------------------------------------------------------------------------------------------
>             |                 | alpha1=0,alpha2=1e-3 | alpha1=1e-6,alpha2=1e-3 | beta=0.1 | beta=0.01 | eps=0.01 | eps=0.1 |
> ------------------------------------------------------------------------------------------------------------------------------------------------------------
>  mean |    75.25   |               75.60                 |                   75.59                   |    75.64    |     75.33     |    75.35    |    75.87  |
>  std     |    0.45      |               0.33                   |                   0.42                     |    0.29      |     0.42       |    0.43      |    0.22    |
> ------------------------------------------------------------------------------------------------------------------------------------------------------------
>
> There are improvements in performance on all three methods.
>
> We don't claim that our regularizer is the state of art that uniformly outperforms all other existing regularizers. Our theoretical result gives justification to any regularizers that effectively control the norm of w^TF(X), which includes our regularizer, weight normalization (Salimans et al., 2016), batch normalization (Ioffe et al., 2015), etc.. The optimal set of hyperparameters depends on the architecture of the model, but for a reasonable choice of hyperparameters (the additional loss introduced by regularizer is comparable with the classification loss) we find our regularizer is always effective on large scale models that tend to overfit.
>
> As mentioned in (Neyshabur et al., 2015), neural networks are equivalent up to some scaling factors passed among layers. As a consequence, there exist unbalanced neural networks with large l2 weights but are equivalent to those with small l2 weights. So the common belief that l2 regularization can "simplify" a model does not necessarily make sense for deep model. One of our main contributions is to interpret the use of l1&l2 regularization in the deep learning setting. We reduce the deep structure into a linear one with MI objective and invertibility of ResNet, and formally justify the use of l2 regularization on both features F(X) and parameters w in the last layer with an interpretation of compressing irrelevant information explained in our proposed information optimization problem.
>
> We derive a theoretically gounded regularizer from our proposed information optimization problem. Our regularizer may seem unusual as it involves feature F(X). We experimentally verify that regularization on F(X) is necessary for deep learning. In addition we observe that naive regularization on w results in a blow up on the norm of F(X), meaning the neural network can absorb the regularization effect by upscaling the feature. We show in Figure 4 that our regularization can hold the magnitude of F(X) while compressing w.
>
> We believe we provide a new perspective to understand regularizations for deep models.
>
>
> 2. We have fixed the citation format in the revised version.

---

> > ### Comment · AnonReviewer1 · 2018-12-04
> > **Reply**
> >
> > It is not clear to me if the proposed regularizer has practical merit, it rather seems like an alternative approach to a more or less well-studied problem. On the other hand, I thank the authors for the updates, the new results indicate the observed effects are consistent with other regularization techniques and across models.
> >
> > I updated my score, but not to clear acceptance due to the aforementioned reason.

---

> > > ### Author Response · Authors · 2018-12-05
> > > **Response to Reviewer 1 Continued**
> > >
> > > We thank the reviewer for the update!
> > >
> > > The main take-away of our work for practitioners should be a design principle for neural networks. We challenge the point of view that the merit of "depth" of neural networks is filtering out irrelevant information layer by layer. The success of ResNet (He et al., 2015) and DenseNet (Huang et al., 2017) clearly conveys the message that neural networks need to preserve more information from the input throughout the layers to perform well.
> > >
> > > However it's somehow counter-intuitive that neural networks that preserve all information (including possibly the irrelevant information) can generalize well. To understand this, we formulate an explicit objective that keeps only the relevant information and take a theoretical approach to draw a connection to the loss objectives that people are familiar with. We see that the product w^TF(X) is crucial for good generalization of invertible neural networks. In addition in Appendix C, we take another point of view to understand why invertibility is beneficial for classification.
> > >
> > > We hope our work clarifies some mysteries of the class of invertible neural networks, which we believe is a good design principle for future work.

---

### Meta-Review · Area_Chair1 · 2018-12-13
**Fascinating perspective with promising initial results, but needs more careful comparison to other regularization methods**

**Confidence:** 4
**Recommendation:** Reject

**Metareview:**

This paper proposes an approach to regularizing classifiers based on invertible networks using concepts from the information bottleneck theory. Because mutual information is invariant under invertible maps, the regularizer only considers the latent representation produced by the last hidden layer in the network and the network parameters that transform that representation into a classification decision. This leads to a combined ℓ1 regularization on the final weights, W, and ℓ2 regularization on W^{T} F(x), where F(x) is the latent representation produced by the last hidden layer. Experiments on CIFAR-100 image classification show that the proposed regularization can improve test performance. The reviewers liked the theoretical analysis, especially proposition 2.1 and its proof, but even after discussion and revision wanted a more careful empirical comparison to established forms of regularization to establish that the proposed approach has practical merit. The authors are encouraged to continue this line of research, building on the fruitful discussions they had with the reviewers.